**Data Availability Statement:** All data needed to replicate this analysis are found in the manuscript and referenced publications.

**Funding:** Dr. Groessl's work on this analysis was partially supported by consulting fees from the

# Cost-effectiveness of Transcendental Meditation (TM) for treating Post-Traumatic Stress Disorder (PTSD)

**Erik J. Groessl** [1,2]*, **Thomas R. Rutledge** [1,3]

**1** VA San Diego Healthcare System, San Diego, CA, United States of America, **2** Herbert Wertheim School of Public Health, University of California San Diego, La Jolla, CA, United States of America, **3** Department of Psychiatry, University of California San Diego, La Jolla, CA, United States of America

* egroessl@ucsd.edu

## Abstract

### Objective

A recent trial found that Transcendental Meditation (TM) was an effective non-trauma focused treatment for veterans with Post-Traumatic Stress Disorder (PTSD). The objective of this analysis was to examine the cost-effectiveness of TM for PTSD based on the trial results.

### Methods

Between 2013–2017, 203 veterans with PTSD were randomized to either TM, Prolonged Exposure (PE), or to a PTSD health education (HE) control. Each group received 12 treatment sessions over 12 weeks. Results indicated that TM was non-inferior to PE for improving PTSD outcomes and both TM and PE were superior to HE, as hypothesized. The proportion of participants with a clinically significant improvement on the CAPS ($\geq$10 point reduction) were TM = 61%, PE = 42%, and HE = 32%. A Markov model was developed to estimate the cost-effectiveness of TM, using the trial effectiveness data. Intervention costs, health care costs, and health utility values associated with response and non-response were derived from scientific literature. Costs were viewed from an organizational perspective and a 5-year time horizon (20 3-month cycles). One-way and probabilistic sensitivity analyses were conducted.

### Results

TM was the dominant treatment strategy over both PE and HE in the cost-effectiveness analysis. TM cost an estimated $1504/12 sessions while PE and HE cost $2,822 and $492, respectively. The higher health care costs associated with non-response to therapy offset intervention cost differences. Findings were robust to variability.

### Conclusion

In summary, using data from a recent RCT, TM was found to both improve health outcomes and reduce total costs in this analysis. Based on these results, further effectiveness trials and wider adoption of TM should be considered.

David Lynch Foundation. The funders had no role in study design, data collection and analysis, decision to publish, or preparation of the manuscript.

**Competing interests:** I have read the journal's policy and the authors of this manuscript have the following competing interests: Erik Groessl, PhD was financially compensated by the David Lynch Foundation for some of his work on this analysis. This non-profit foundation promotes Transcendental Meditation (TM) as a beneficial health practice and charges a fee for training in TM. This does not alter our adherence to PLOS ONE policies on sharing data and materials.

## Introduction

Posttraumatic stress disorder (PTSD) is a medical diagnosis used to describe a chronic condition in which these trauma-related symptoms do not diminish substantially and become functionally impairing [1]. PTSD has broad health effects, including higher rates of depression [1], substance use [2], suicidality [3], physical health problems [4], disability [5], and chronic pain [6]. PTSD disproportionately afflicts US Veterans, resulting in an enormous burden on Veterans' mental health, physical health, functional well-being, and health care resources nationally and internationally.

Evidence-based treatments in the most recent PTSD treatment guidelines include trauma-focused, cognitive behavioral therapies such as prolonged exposure (PE) and cognitive processing therapy (CPT) [7, 8]. Although trauma-focused therapies are generally more effective than non-trauma focused therapies [9], trauma-focused treatments are not effective for all persons with PTSD because many either refuse or do not complete these therapies because they require the aversive experience of focusing on the trauma itself [10]. Complementary and integrative health treatments (CIH) are a promising alternative treatment to these therapies because they have been shown to generally be safe, have few side effects, and are non-trauma focused. Among CIH treatments, Transcendental Meditation (TM) is an evidence-based, non-trauma-focused PTSD treatment that involves the use of a mantra (sound), without concentration or contemplation. The effects of TM as a health intervention have been documented in numerous studies [11–16].

A US Department of Defense (DOD) funded non-inferiority clinical trial randomized 203 military veterans with PTSD from the VA San Diego Healthcare System to either Transcendental Meditation™, prolonged exposure therapy (PE), or an active PTSD health education (HE) control group [17]. The primary findings of this study indicated that TM was non-inferior to PE on change in PTSD symptoms from baseline to the end of the 12-week intervention [17], using scientifically valid measures of PTSD in military settings (Clinician-Administered PTSD Scale (CAPS) and Posttraumatic Stress Disorder Checklist-military (PCL-M)).

This study was important for a number of reasons. First, although effective for many persons with PTSD when fully delivered, many persons undergoing PE do not complete treatment or do not respond, likely because of adversity to the trauma-focused nature of PE therapy. Thus, when shown to be effective, non-trauma-focused therapies like TM offer an alternative for those unable or unwilling to tolerate the full PE treatment. Having additional non-trauma focused options can potentially increase the total proportion of persons with PTSD achieving clinical benefit. Secondly, health care costs in the US are higher per capita than anywhere else in the world and continue to increase. Thus, it is important for researchers to consider and examine the costs of delivering new therapies throughout all phases of research. Once a therapy like TM has been shown to be effective, it is crucial to ask "at what cost" are the incremental benefits obtained, especially in comparison to real-world gold standard options like PE.

Therefore, using data from the above clinical trial supporting the efficacy of TM for reducing PTSD symptoms, the current paper examined the costs of TM and incremental cost-effectiveness of TM compared with a guideline-based therapy. No previous studies of the cost-effectiveness of TM for PTSD have been published.

## Materials and methods

### Study design

Effectiveness data were provided by a published randomized clinical trial [17]. The analytic methods followed the recommendations of the 2^nd Panel on Cost-Effectiveness in Health and

Medicine [18, 19] and the Consolidated Health Economic Evaluation Reporting Standards (CHEERS) statement [20]. A US health care system perspective was assumed for the analysis, and inflation-adjusted costs were calculated in 2023 US dollars. The incremental cost/QALY and the incremental cost/ clinical improvement in PTSD were calculated between the three interventions noted above, HE, TM, and PE. The analysis used only published or publicly available aggregate data and did not use health data from individuals. Therefore, human subjects approval was not required.

## Model structure

A Markov model was created in which participants received one of three treatments from the clinical trial described above. The model cycle duration of three months was chosen to align with the trial outcomes assessment period. The possible treatment outcomes after the initial cycle were clinical improvement (CI), no clinical improvement (NI), or death (all cause). Subsequent cycles consisted of the same outcome options each cycle, including the ability to transition from CI or NI to one of the other outcomes (Fig 1). A total of 20 3-month cycles were modeled, representing a 5-year time horizon. was chosen because of the limited length of follow-up data from the clinical trial, yet also aligns with the period modeled in a recent cost-effectiveness analysis of a PTSD treatment [21]. An annual discount of 3% was applied to costs and QALYs accumulating in cycles 5 to 20. The analysis was conducted using TreeAge Pro Healthcare 2024 software [22].

## Effectiveness data

The most clinically relevant measure of effectiveness from the TM trial was the percentage of participants with a clinically significant improvement, defined as a ≥10-point reduction in CAPS score at the 12-week post-treatment assessment. This cutoff was previously established [23] and remains a more conservative definition than a more recent recommended cutoff of ≥8 point reduction in CAPS [24]. The percentages of participants meeting the ≥10 point reduction criterion were TM = 61%, PE = 42%, and HE = 32% [17]. To account for clinical improvements that may have been temporary, we estimated that 6% of clinically improved participants annually reverted back to not improved status annually [25]. Likewise, to account for non-responders seeking other PTSD treatments after the 12-week study or spontaneous remission, we estimated that 10% of non-responders improved annually. In the absence of

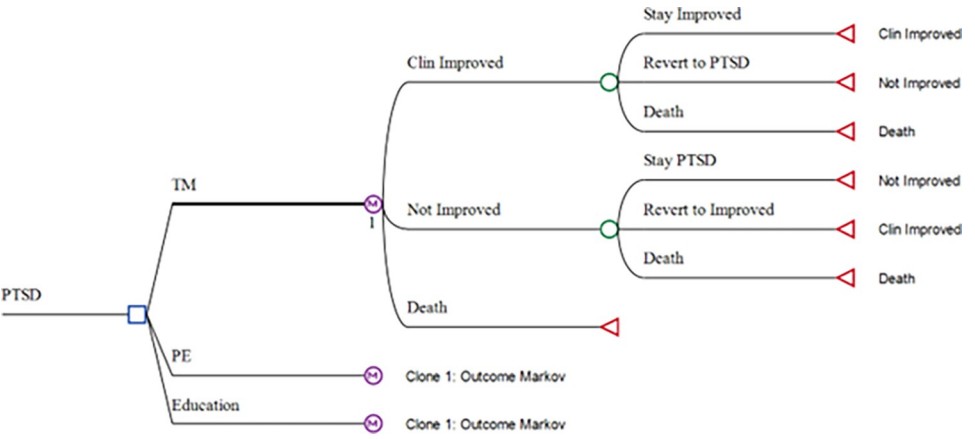

**Fig 1. Markov model structure.**

data on PTSD, this estimate was based on a study of untreated depression [26]. No evidence was found to suggest that rates of relapse or emission would differ by group.

## Mortality rates

Based on the Nidich et al. trial sample in which 83% of participants were male, and the mean age was 47 years [17], a mortality rate was calculated from the most recently available United States Life Tables, 2020 [27]. The annual mortality rate of 0.0045 was used for clinically improved participants. Data from multiple studies indicates that mortality rates are elevated in persons with PTSD, with one study finding a more than two-fold increase in mortality after adjusting for cardiovascular factors [28]. However, a more recent meta-analytic study that included data from 30 studies and over 2 million participants found a smaller effect (RR = 1.32) [29]. Thus, we adjusted the general population mortality rate by 1.32 for those the no clinical improvement in PTSD health state to 0.0059.

## Health utility values

Estimates of health utility values for persons with and without PTSD were obtained from the scientific literature. A recent cost-effectiveness analysis used multiple tiers of PTSD severity with utility values ranging from 0.37 for "extreme" PTSD to 0.90 for a PTSD asymptomatic person [21]. Although the severity of PTSD in the Nidich trial appears to fit with the "severe" and "extreme" tiers used by Marseille et al., all tiers could not be replicated and the range of utilities used seemed larger than other studies. Thus, to be more conservative and straightforward, we used more conservative estimates from a recent high-quality, meta-analytic approach to CEA in persons with PTSD. With the Nidich sample being 90% male, we used values of 0.54 for those with PTSD and 0.63 for those who reported a clinically significant improvement in PTSD [30].

## Intervention costs

The TM intervention consisted of 1 longer, 90-minute introductory session, and eleven 60-minute follow-up sessions, all delivered by authorized trained personnel. More details are available in the main results paper [17]. The actual cost paid by the study to TM for providing the TM intervention for 12 weeks was $1,450/participant. This cost was approximately 50% greater than the retail cost quoted at the TM website [31] because the study intervention consisted of additional weekly follow-up sessions in order to match the guideline-recommend length of PE therapy for PTSD. Beyond the intervention instructional fee, delivery of the TM intervention also required some support from an administrative person to schedule TM sessions and send reminders. Using the US Labor Bureau data [32] we estimated that a Healthcare Support Worker with a mean hourly wage of $19.24 would spend up to 10 minutes per session, or 2 hours per participant total across the 12 sessions on scheduling and reminders. We then added 69% overhead costs to the personnel total, accounting for facilities costs and other typical indirect costs associated with running an outpatient healthcare program [33]. Overhead was not applied to the TM instructor cost since this was an inclusive external service. Thus, as shown in Table 1 below, the total estimated costs of TM was $1,504 per participant.

To estimate the cost of delivering the PE intervention, we reviewed scientific literature for cost estimates of PE for PTSD or other conditions We derived our PE cost from a more recent in-depth cost-effectiveness analysis conducted in the UK [30]. This study estimated nine 90-minutes individual sessions of trauma-focused CBT for PTSD delivered by a licensed psychologist to cost £1,368 in 2017 currency. We converted this to US dollars ($1,732), and then to 12 90-minute sessions ($2,309) and finally to year 2023 US$ using the Bureau of Labor

Statistics CPI calculator [34]. This produced a final estimate of $2,822/person for the PE intervention.

For the health education intervention, a CEA from 2016 found that the cost of 32 group health education sessions for older adults to be $1,001 per participant, or about $31/participant per session [35]. This amount includes administrative costs and overhead. With 12 sessions being delivered in the Nidich study [17], this comes to an estimated $375 /participant in 2013 US dollars. Adjusting for inflation [34], the health education is estimated to cost $492/participant in 2023 US dollars.

### Health care costs

Estimated annual health care costs for our model were derived from an analysis published by Harper et al (2022) [36] in which costs for VA patients with and without PTSD were compared. This analysis adjusted costs for multiple confounding factors including past usage patterns and comorbidities and found that health care costs remained approximately 55% higher in those with PTSD after adjustments. This resulted in annual health care costs of $9,109 for individuals with PTSD and $5,887 for those without PTSD. Other studies that include data from non-Veteran sources suggest that the 55% proportional increase may be a conservative estimate [37, 38]. Costs were then adjusted for inflation from 2012 to 2023 US Dollars, bringing the 2023 annual estimates to $12,154 for individuals with PTSD and $7,855 for those without PTSD. The estimates provided by Harper had little variability, with small standard deviations that are atypical of health care costs data.

### Sensitivity analyses

Univariate, one-way sensitivity analyses were performed using TreeAge software. A multivariate probabilistic sensitivity analysis was also conducted to examine the impact of variance on costs, QALYs, and NMB across multiple input parameters simultaneously. Variability estimates for utilities were obtained from previous studies [30, 39]. The probability of clinical improvement was varied using 95% confidence intervals from the Nidich et al trial [17]. Intervention cost and health care costs estimates were varied 20% in each direction. Input estimates for mortality rates were based on the US Life Tables and a large meta-analysis of over 2 million participants and were thus varied 10% in each direction.

Our one-way sensitivity analyses display model outputs in units of incremental net monetary benefit (NMB), which integrates QALYs and the willingness-to-pay (WTP) function [18, 40]. The incremental NMB is especially useful when treatments being studied are dominant or dominated [41]. NMB converts QALYs to dollars using the chosen WTP rate, and thus represents the amount of additional monetary benefit per person provided by the dominant strategy given the WTP of $50,000. A WTP threshold of $50,000 has been a commonly used threshold in cost-effectiveness analysis [40, 42] and is currently considered conservative since it has not been adjusted for inflation.

## Results

The main results of the incremental cost-effectiveness analysis are presented in Table 2. Across the 5-year time horizon, an estimated 2.66, 2.75, and 2.69 QALYs accumulated in the HE, TM, and PE groups, respectively. The incremental differences of 0.06 (TM vs. PE) and 0.09 QALYs (TM vs. HE) meaningful and reflect the RCT findings that if 100 Veterans with PTSD were assigned to each group, the TM group would result in 19 and 29 more participants with clinical improvements, when compared to the PE and HE groups, respectively.

**Table 1. Markov model inputs.**

| Parameters | Distribution | Base-case | Low | High | Reference/ Source |
|---|---|---|---|---|---|
| Effect inputs | | | | | |
| Health Education | Beta | 0.32 | 0.256 | 0.384 | [17] |
| Transcendental Meditation | Beta | 0.61 | 0.490 | 0.730 | [17] |
| Prolonged Exposure | Beta | 0.42 | 0.336 | 0.504 | [17] |
| Clinical inputs | | | | | |
| Age at baseline | Fixed | 47 | | | [17] |
| Gender–Male | Fixed | 83% | | | [17] |
| Relapse rate | Fixed | 6% | | | [25] |
| Remission rate | Fixed | 10% | | | [26] |
| Mortality rate - clinically improved (annual) | Beta | 0.0045 | 0.0036 | .0054 | [34] |
| Mortality rate - not clinically improved (annual) | Beta | 0.0059 | .0047 | .0071 | [29, 34] |
| Cost inputs, $ | | | | | |
| Health Education | Triangular | $492 | $394 | $590 | [35] |
| Transcendental Meditation | Triangular | $1,504 | $1,203 | $1,805 | Methods section |
| Prolonged Exposure | Triangular | $2,822 | $1,605 | $3,386 | [30] |
| Health care costs—clinically improved (annual) | Gamma | $7,855 | $6,284 | $9,426 | [36] |
| Health care costs – not clinically improved (annual) | Gamma | $12,154 | $9,723 | $14,585 | [36] |
| Utility inputs | | | | | |
| Clinically Improved PTSD | Beta | 0.63 | 0.50 | 0.76 | [30, 39] |
| Not Clinically Improved PTSD | Beta | 0.54 | 0.43 | 0.65 | [30, 39] |
| Policy inputs | | | | | |
| Time horizon | 5 years | | | | Methods section |
| Discount rate | 3% | | | | [18] |

When only considering the intervention cost and the effects found in the trial, dividing the intervention cost per participant by the rates of clinically improved patients provided a base estimate of intervention cost/improved patient for each group. Before including health care costs, the intervention cost/improved patient for each group amounts to $1,538 for HE, $2,466 for TM, and $6,719 for PE. As expected, the HE control was the least costly intervention yet also was the least effective. When including total health care costs from a healthcare organization perspective, a different picture emerges. Total costs from the 5-year model, consisting of intervention costs and subsequent health care costs, were $45,679, $42,982, and $46,730 for HE, TM, and PE respectively. Thus, lower subsequent health care costs associated with more effective treatments offsets much (PE) or all (TM) of the higher initial cost of delivering those treatments.

When examining the incremental cost-effectiveness ratios (ICERs), the most important comparison is between TM and PE because PE can be considered a gold standard treatment option while TM is the new treatment option of interest. As shown in the highlighted section of Table 2, our analysis indicates that overall, TM costs $3,748 less per person than PE over a 5-year interval, and produced more health benefit than PE. This was true using both health effectiveness measures, QALYs and proportion of participants with a clinical improvement. Thus, an incremental cost-effectiveness ration cannot be calculated. TM was also less expensive overall and more effective than HE as expected. Thus, TM is considered the dominant

**Table 2. Incremental cost-effectiveness results.**

| | HE | TM | PE | Difference TM vs HE | Difference TM vs PE | Difference PE vs HE |
|---|---|---|---|---|---|---|
| **Effectiveness** | | | | | | |
| •rate of clinical improvement | 32% | 61% | 42% | Δ29% | Δ19% | Δ10% |
| •QALYs/person | 2.66 | 2.75 | 2.69 | Δ0.09 | Δ0.06 | Δ0.03 |
| **Costs** | | | | | | |
| •Intervention costs/person | $492 | $1,504 | $2,822 | $1,012 | -$1,318 | $2,330 |
| •Health care costs/person over 5-yr analysis | $45,187 | $41,478 | $43,931 | -$3,709 | -$2,453 | -$1,256 |
| **Total Costs/person** | **$45,679** | **$42,982** | **$46,730** | **-$2,697** | **-$3,748** | **$1,051** |
| **ICER ($ cost/QALY)** | | | | dominant | dominant | $35,033 |
| **Incremental NMB** | | | | **$7,136** | **$6,656** | **$480** |

HE = Health Education; TM = Transcendental Meditation; PE = Prolonged Exposure Therapy

ICER = Incremental Cost-Effectiveness Ratio

NMB = Net Monetary Benefit

Dominant–refers to an intervention that both costs less and provides more health improvement

treatment choice in these comparative analyses. The NMB of $6,656 reflects the monetary value of the added health effects assuming a WTP f $50,000/QALY.

A secondary comparison between PE and HE indicated that PE resulted in slightly higher total costs yet was a more effective treatment. This comparison resulted in an ICER of $35,033/ QALY, or $10,510/ additional clinically improved participant.

To examine the impact of assumptions and estimated input parameters on our main results in which TM is compared with PE, we initially conducted one-way sensitivity analyses. As shown in Fig 2, the health utility rates for persons with PTSD and for those with clinically improved PTSD had relatively larger impacts on the NMB results. The probability of improvement for TM and PE participants derived from the Nidich RCT also had larger impacts on NMB than most other variables. However, as shown, it was clear that overall incremental NMB remained positive regardless of the variations in any one variable. None of the error bars approach an NMB of 0.

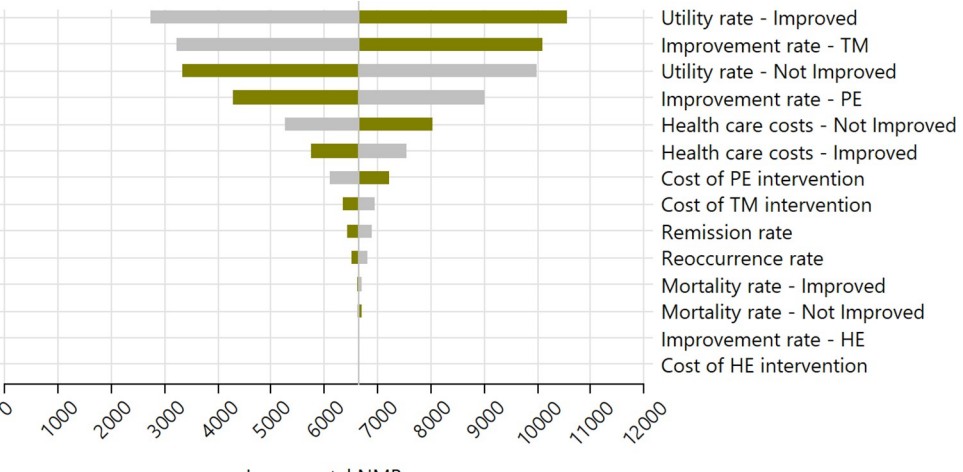

**Fig 2. One-way sensitivity Tornado Chart.** Variation in incremental net monetary benefit ($).

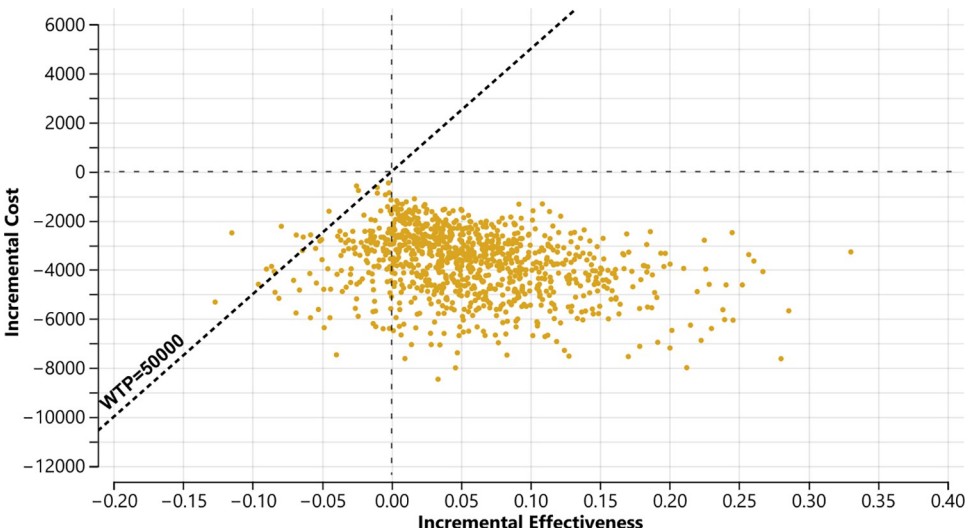

**Fig 3. Incremental cost-effectiveness scatterplot from probabilistic sensitivity analysis (TM vs PE).**

To examine the multivariate sensitivity of our results by varying all input parameters along the sampling distributions listed in Table 1, we conducted a probabilistic sensitivity analysis with all three treatments, TM was the optimal strategy in 96% of the scenarios generated. Our main focus, however, was on comparing TM to PE which was the primary comparison of interest in the clinical trial as noted above. The results of 1000 simulations are presented in the scatterplot in Fig 3. When varying multiple relevant parameters at the same time, TM was the optimal strategy and dominated PE in 87% of the scenarios generated, being more effective and less costly. There were some scenarios (12%) in which PE was estimated to be slightly more effective, but the higher cost of PE resulted in the ICER being above $50,000/QALY. The acceptability curve (S1 Fig) indicated that TM's dominance as the optimal choice was not sensitive to the choice of WTP threshold.

## Discussion

Recent research has demonstrated that mind-body interventions such as yoga and meditation have promise as non-trauma focused treatment options for PTSD. A number of pilot studies and one large RCT show benefits of yoga for persons with PTSD [43, 44]. Other evidence supports the use of meditation interventions for treating PTSD [45, 46]. In addition to the safety and efficacy data provided by these latter studies, the current paper offers novel findings regarding the potential cost-effectiveness of transcendental meditation based on a recent randomized controlled trial. The results of our analysis are based on a recent full-scale RCT of TM for PTSD in military veterans receiving care from a VA medical center [17].

Our main results indicate that TM had the largest benefit among the three treatment groups, despite costing less than Prolonged Exposure, an evidence-based trauma-focused PTSD treatment. Given that PTSD has been associated with higher health care costs in multiple studies [36–38], our decision model analysis estimated that lower subsequent health care costs in the 5 years after treatment can more than offset intervention costs from a healthcare organization perspective. Sensitivity analyses suggest that these finding are robust to variability in estimated model parameters. The results indicate that TM appears to be highly cost-effective, and there is "strong evidence for adoption" [40, 47]. Established ratings of PTSD severity in this study indicate that participants were experiencing very high levels of PTSD at baseline

and over 60% of participants in the TM group reported a clinically significant improvement. Clinically, it is important to consider that our effectiveness findings included treatment response irrespective of whether a participant completed therapy. All randomized participants in the prior RCT were included in analyses. The difference in dropout (completion of less than 8 of the 12 sessions) rates was not significant as reported previously (25% TM vs 38% PE) [17], but it may have contributed to the overall response rate. A secondary finding of the current study was that both TM and PE were found to be more effective and to ultimately cost less than a health education control condition after 5 years.

When interpreting the finding that when compared to PE, TM saved an estimated $3,748 per person over 5 years, it is important to remember that this figure represents the average cost savings when including all participants randomized to each condition, not just those that responded. Thus, the benefits are expected to be multiplicative, meaning that if 100 veterans with PTSD were treated with TM instead of PE, the expected savings would be about $375,000 based on data from this trial. Expanding the sample and treating 100,000 veterans with PTSD instead of PE would be expected to save $375 million over 5 years.

Our findings align with at least one prior study of the impact of TM on health care costs [48]. In that analysis, researchers used medical claims data to show that among high-cost health system users, "physician payment costs" were lower over a 5-year period among participants that started TM than in those that did not. There were no differences in health care costs prior to starting TM. However, broader health care costs beyond physician payments were not available, and the overall study design warrants tentative conclusions. Despite a lack of studies on the cost-effectiveness of mind-body interventions for treating PTSD, a number of studies have shown that meditation and yoga can be effective options for treating and reducing symptoms of PTSD [44, 45, 49, 50]. Given those findings, evidence that PTSD results in higher health care costs, and the relatively low cost of delivering mind-body interventions, it is not surprising that mind-body interventions are likely a cost-effective treatment for PTSD [51].

Considering the health economic impact of more traditional psychological treatments for PTSD, one systematic review found that across 30 different studies conducted at that time, good methodology was generally lacking for health economic analyses [52]. However, the same researchers found there was sufficient evidence to suggest that PTSD resulted in significantly higher costs in terms of health care [53], as well as societal costs from lost productivity and indirect impacts. More recently, a decision-analytic modeling approach ranked 10 different psychological treatments for PTSD [30]. Eye movement desensitization and reprocessing (EMDR) ranked as most cost-effective followed by combined somatic/cognitive therapies.

When thinking about next steps it is important to weigh the confidence conveyed in the results presented herein. The input variables that most affected our results were the clinical effectiveness rates and the health utility assigned to PTSD vs clinical improvement. The utility values were obtained from published literature and were chosen as conservative options, representing a 0.09 difference between PTSD and clinical improvement. The effectiveness results were based on a single RCT, but that study was well-designed and appears rigorous. Confirming the effectiveness of TM for PTSD in one or more additional studies is important, but currently there is no reason to doubt the overall conclusions of our analysis. After varying these inputs 20% bidirectionally, TM remained less costly and more effective. Importantly, the probabilistic sensitivity analysis confirmed our conclusions with TM being the optimal choice in 86% of the scenarios generated. In summary, this is the first analysis of the cost-effectiveness of TM for PTSD and uses the best available estimates, yet conclusions can remain tentative until more evidence regarding the effectiveness of TM is available.

Overall, the results add potentially valuable information to the clinical decision-making process. Until more data is available, there is no indication that TM should replace PE as a

leading treatment for PTSD. However, it should be considered as a cost-effective treatment option, either as the initial treatment option, or secondarily, for anyone that was unable to participate in or complete PE or did not respond. Although military veterans with PTSD preferred PE over medications in a study published in 2014, they were aware of very few other alternatives [54]. Other data show that people with PTSD were more likely to be "disinclined" to choose PE over other treatment options, and having options was of significant value [55]. TM provides an additional non-trauma focused treatment option.

## Limitations

The analyses presented herein have a number of limitations. Our analyses, results, and conclusions are based on data from a single RCT [17]. That RCT was rigorously designed and found moderate to large effects, but confirmatory studies are necessary to strengthen conclusions. Notably, that RCT was also conducted with a sample of mostly male US military veterans with PTSD. Thus, the prior RCT and the cost-effectiveness results presented here may not generalized to non-veterans with PTSD. Further research should be conducted on the efficacy and cost-effectiveness of TM in broader PTSD samples.

It is a limitation that the RCT on which our analysis is based did not track participant health care utilization, so health care costs were estimated in our analysis. Finally, the RCT by Nidich et al. was limited to outcomes measured at three months.

To offset these limitations and bolster confidence in the analysis results, estimates of model inputs were kept conservative. We used conservative estimates of the impact of PTSD on mortality, health care costs, and health utility when multiple estimates were available. We estimated that 10% of improved patients would revert back to PTSD annually and refrained from estimating any spontaneous remission of PTSD.

## Conclusion

In conclusion, there is growing evidence supporting the efficacy–and now cost-effectiveness– of mind-body interventions such as TM for treating PTSD in veterans with moderate to severe symptoms.

## Supporting information

**S1 Checklist. CHEERS 2022 checklist.**
(PDF)

**S1 Fig. Willingness-to-pay acceptability curve.**
(BMP)

## Author Contributions

**Conceptualization:** Erik J. Groessl.

**Data curation:** Thomas R. Rutledge.

**Formal analysis:** Erik J. Groessl.

**Methodology:** Erik J. Groessl.

**Validation:** Thomas R. Rutledge.

**Visualization:** Erik J. Groessl.

**Writing – original draft:** Erik J. Groessl, Thomas R. Rutledge.

**Writing – review & editing:** Erik J. Groessl, Thomas R. Rutledge.

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
