## [Decision Letter · Decision Letter 0]

5 Nov 2024

PONE-D-24-32697Cost-Effectiveness of Transcendental Meditation (TM) for Treating Post-Traumatic Stress Disorder (PTSD)PLOS ONE

Dear Dr. Groessl,

Thank you for submitting your manuscript to PLOS ONE. After careful consideration, we feel that it has merit but does not fully meet PLOS ONE’s publication criteria as it currently stands. Therefore, we invite you to submit a revised version of the manuscript that addresses the points raised during the review process.

We look forward to receiving your revised manuscript.

Kind regards,

Ben Green

Academic Editor

PLOS ONE

“As noted in the manuscript Acknowledgments section, Erik Groessl was financially compensated by the David Lynch Foundation for some of his work on this analysis.”

“Dr. Groessl’s work on this analysis was partially supported by consulting fees from the David Lynch Foundation.

Dr. Groessl and Dr. Rutledge have received research support for other studies from the David Lynch Foundation.”

“As noted in the manuscript Acknowledgments section, Erik Groessl was financially compensated by the David Lynch Foundation for some of his work on this analysis.”

“I have read the journal's policy and the authors of this manuscript have the following competing interests: Erik Groessl, PhD was financially compensated by the David Lynch Foundation for some of his work on this analysis. This non-profit foundation promotes Transcendental Meditation (TM) as a beneficial health practice and charges a fee for training in TM.”

5. In the online submission form you indicate that your data is not available for proprietary reasons and have provided a contact point for accessing this data. Please note that your current contact point is a co-author on this manuscript. According to our Data Policy, the contact point must not be an author on the manuscript and must be an institutional contact, ideally not an individual. Please revise your data statement to a non-author institutional point of contact, such as a data access or ethics committee, and send this to us via return email. Please also include contact information for the third party organization, and please include the full citation of where the data can be found.

Reviewers' comments:

Reviewer's Responses to Questions

**Comments to the Author**

1. Is the manuscript technically sound, and do the data support the conclusions?

Reviewer #1: Yes

Reviewer #2: Yes

2. Has the statistical analysis been performed appropriately and rigorously? 

Reviewer #1: No

Reviewer #2: Yes

3. Have the authors made all data underlying the findings in their manuscript fully available?

Reviewer #1: Yes

Reviewer #2: Yes

4. Is the manuscript presented in an intelligible fashion and written in standard English?

Reviewer #1: Yes

Reviewer #2: Yes

5. Review Comments to the Author

Reviewer #1: 1. Although the article explores the cost-effectiveness of TM, the introduction section fails to fully explain the potential impact and application value of this study in practical clinical settings. The lack of a more specific explanation of the importance of the research results in readers not having a deep understanding of the practical significance of the research.

2. Lack of treatment compliance assessment: The article did not mention how to address patients' compliance issues during treatment, especially for treatment methods such as TM and PE that require continuous patient participation. Whether patients can fully follow the treatment plan for meditation or exposure therapy may significantly affect the efficacy. The results section did not discuss how patient treatment compliance affects the cost-effectiveness analysis results. If patients are unable to continue TM or PE, the efficacy may be greatly reduced, which will directly affect the economic conclusions of the study.

3. The statistical analysis section of the article is relatively simple, and it is recommended to further refine it. For example, adding descriptions of multivariate regression analysis or other multivariate analyses to explain the combined impact of different variables on the results.

4. Improve chart and data display, add more intuitive presentation methods, so that readers can easily understand the results of cost-benefit analysis.

5. The conclusion of the article points out that TM is more cost-effective than PE and HE in PTSD treatment, but due to the limitations of the study sample (only veterans), this conclusion may be difficult to extrapolate to other populations. Especially for non veterans, PTSD patients of different age groups and genders, they may have different responses to TM. It is suggested to point out that this conclusion is mainly applicable to the veteran population and call for further verification in other PTSD patient populations. Discuss how to optimize the mental health policies for veterans through this study, and how to promote TM as an effective treatment for PTSD.

6. The discussion section on the comparison between Transcendental Meditation (TM), Exposure Therapy (PE), and Health Education (HE) is more superficial and focuses more on the cost-effectiveness results. There is no in-depth analysis of the advantages and disadvantages of these therapies in terms of clinical improvement.

Reviewer #2: This study aimed to examine the costs of Transcendental Meditation and incremental cost-effectiveness of Transcendental Meditation compared with a guideline-based therapy. The strength of this study was to conduct analysis using a rigorous method. However, there were some concerns in this study.

First, in the Introduction, the authors had better cite the following Cochrane review and describe the article: “Bisson JI, Roberts NP, Andrew M, Cooper R, Lewis C. Psychological therapies for chronic post‐traumatic stress disorder (PTSD) in adults. Cochrane Database of Systematic Reviews 2013, Issue 12. Art. No.: CD003388. DOI: 10.1002/14651858.CD003388.pub4.”

Second, in the Materials and Methods, they had better cite Consolidated Health Economic Evaluation Reporting Standards 2022 (CHEERS 2022) statement and adhere to the statement instead of the previous version of CHEERS statement.

Reference

Husereau, D., Drummond, M., Augustovski, F. et al. Consolidated Health Economic Evaluation Reporting Standards 2022 (CHEERS 2022) statement: updated reporting guidance for health economic evaluations. BMC Med 20, 23 (2022). https://doi.org/10.1186/s12916-021-02204-0

Third, they had better add CHEERS 2022 Checklist as a supplementary file.

The statements in the paper corresponding to each item in this checklist can be added to https://don-husereau.shinyapps.io/CHEERS/ and this checklist can be downloaded.

I have cited the explanation to use the checklist as follows:

“Those using the checklist should indicate the section of the manuscript where relevant information can be found. We recommend using a section heading with a paragraph number since referring to line or page numbers becomes confusing as repagination or line number changes occur within or after the publication process. If an item does not apply to a particular economic evaluation (e.g., items 11-13 for cost analyses, or items 16 and 22 for non-modelling studies), then checklist users are encouraged to report “Not Applicable”. If information is otherwise not reported, checklist users are encouraged to write, “Not Reported”. Users should avoid the term “Not conducted” as CHEERS is intended to guide and capture reporting.”

Fourth, they had better revise the manuscript, or add new limitations in the discussion if any important items in the CHEERS 2022 Checklist were not met.

6. PLOS authors have the option to publish the peer review history of their article (what does this mean?). If published, this will include your full peer review and any attached files.

Reviewer #1: No

Reviewer #2: **Yes: **Masahiro Banno

---

## [Author Response · Author response to Decision Letter 0]

17 Dec 2024

Reviewer #1: 1. Although the article explores the cost-effectiveness of TM, the introduction section fails to fully explain the potential impact and application value of this study in practical clinical settings. The lack of a more specific explanation of the importance of the research results in readers not having a deep understanding of the practical significance of the research.

Thank you for this important suggestion. We have added a solid paragraph, with references, to the introduction (lines 80-90) to address this point and better frame the research question for the audience.

2. Lack of treatment compliance assessment: The article did not mention how to address patients' compliance issues during treatment, especially for treatment methods such as TM and PE that require continuous patient participation. Whether patients can fully follow the treatment plan for meditation or exposure therapy may significantly affect the efficacy. The results section did not discuss how patient treatment compliance affects the cost-effectiveness analysis results. If patients are unable to continue TM or PE, the efficacy may be greatly reduced, which will directly affect the economic conclusions of the study.

We have added a few sentences (lines 308-313) to highlight this issue, and explain that treatment dropout rates may have contributed to difference in effectiveness and therefore, cost-effectiveness. However, compliance rates were previously reported in the main results paper of the RCT, and our analysis only focuses on cost-effectiveness. Thus, since those rates were not significantly different in the prior study, we did not want to overemphasize that point. 

3. The statistical analysis section of the article is relatively simple, and it is recommended to further refine it. For example, adding descriptions of multivariate regression analysis or other multivariate analyses to explain the combined impact of different variables on the results.

We have clarified that probabilistic sensitivity analysis is a multivariate analysis of the impact of all input parameters simultaneously (line 206). We have also added a paragraph towards the end of the discussion to better discuss possible influences of model inputs along with the overall conclusions (lines 346-358). We also appreciate the suggestion of an econometric approach. However, because the prior study did not collect healthcare cost data or QALY data from individual subjects, we were not able to reliably calculate individual cost-effectiveness estimates. Limiting our options for explanatory analyses. 

4. Improve chart and data display, add more intuitive presentation methods, so that readers can easily understand the results of cost-benefit analysis.

We agree that the data presentation may not have been clear to those less familiar with cost-effectiveness analyses because there were 3 intervention groups and 2 different measures of effectiveness (QALYs and rate of clinical improvement). Thus, we have edited Table 2 (lines 563-568) to provide more information, explain abbreviations, and highlight the most important findings in light blue. We have also rewritten the text (lines 244-263) describing these results to be clearer and to focus the attention on the highlighted section and its importance. Although we attempted, we did not find a better way to present these data in a single clear figure or display.

5. The conclusion of the article points out that TM is more cost-effective than PE and HE in PTSD treatment, but due to the limitations of the study sample (only veterans), this conclusion may be difficult to extrapolate to other populations. Especially for non-veterans, PTSD patients of different age groups and genders, they may have different responses to TM. It is suggested to point out that this conclusion is mainly applicable to the veteran population and call for further verification in other PTSD patient populations. Discuss how to optimize the mental health policies for veterans through this study, and how to promote TM as an effective treatment for PTSD.

We agree with this suggestion, and we have expanded our limitations section (lines 371-378) as recommended. 

6. The discussion section on the comparison between Transcendental Meditation (TM), Exposure Therapy (PE), and Health Education (HE) is more superficial and focuses more on the cost-effectiveness results. There is no in-depth analysis of the advantages and disadvantages of these therapies in terms of clinical improvement.

Thank you. We have added two solid paragraphs (lines 346-368) towards the end of the discussion to better discuss possible influences of model inputs, along with the overall conclusions, and implications for clinical decision-making. Because, our paper is focused on health economics, an in-depth, comprehensive discussion of PTSD clinical decision-making seems beyond the current scope. Additionally, as we now better note in the limitations section, there remains just one full-scale RCT of TM for PTSD and this is thus the first cost-effectiveness analysis. Given these limitations, we think it is better to keep conclusions somewhat tentative and not speculate or over interpret results. That is why we kept the discussion more focused.

Reviewer #2: This study aimed to examine the costs of Transcendental Meditation and incremental cost-effectiveness of Transcendental Meditation compared with a guideline-based therapy. The strength of this study was to conduct analysis using a rigorous method. However, there were some concerns in this study.

First, in the Introduction, the authors had better cite the following Cochrane review and describe the article: “Bisson JI, Roberts NP, Andrew M, Cooper R, Lewis C. Psychological therapies for chronic post‐traumatic stress disorder (PTSD) in adults. Cochrane Database of Systematic Reviews 2013, Issue 12. Art. No.: CD003388. DOI: 10.1002/14651858.CD003388.pub4.”

Thank you for this suggestion. We agree this is an important review and we now cite this in the 2nd paragraph of the introduction (lines 60-64). 

Second, in the Materials and Methods, they had better cite Consolidated Health Economic Evaluation Reporting Standards 2022 (CHEERS 2022) statement and adhere to the statement instead of the previous version of CHEERS statement.

Reference

Husereau, D., Drummond, M., Augustovski, F. et al. Consolidated Health Economic Evaluation Reporting Standards 2022 (CHEERS 2022) statement: updated reporting guidance for health economic evaluations. BMC Med 20, 23 (2022). https://doi.org/10.1186/s12916-021-02204-0.

We have updated the manuscript (line 105). Thank you for catching this error. We had this reference on file and intended to cite the CHEERS 2022 standards. 

Third, they had better add CHEERS 2022 Checklist as a supplementary file.

The statements in the paper corresponding to each item in this checklist can be added to https://don-husereau.shinyapps.io/CHEERS/ and this checklist can be downloaded.

We have completed the CHEERS 2022 checklist and added this as a supplementary file. (S2 CHEERS checklist)

Fourth, they had better revise the manuscript, or add new limitations in the discussion if any important items in the CHEERS 2022 Checklist were not met.

All items were met except for 3-4 checklist items that were not applicable. These pertained only to subgroup analyses which were not performed, and to stakeholder input which was not obtained for this study. The funder had no input for the analysis.

---

## [Editor Report · Decision Letter 1]

20 Dec 2024

Cost-effectiveness of Transcendental Meditation (TM) for treating Post-traumatic stress disorder (PTSD)

PONE-D-24-32697R1

Dear Dr. Groessl,

We’re pleased to inform you that your manuscript has been judged scientifically suitable for publication and will be formally accepted for publication once it meets all outstanding technical requirements.

Kind regards,

Ben Green

Academic Editor

PLOS ONE
---

## [Editor Report · Acceptance letter]

15 Jan 2025

PONE-D-24-32697R1 

PLOS ONE

Dear Dr. Groessl, 

I'm pleased to inform you that your manuscript has been deemed suitable for publication in PLOS ONE. Congratulations! Your manuscript is now being handed over to our production team.

Kind regards, 

on behalf of

Professor Ben Green 

Academic Editor

PLOS ONE